# Diagnosis of Septic Body Cavity Effusion in Dogs and Cats: Cytology vs. Bacterial Culture

**DOI:** 10.3390/ani14121762

**Published:** 2024-06-11

**Authors:** Marta Medardo, Paolo Capozza, Walter Bertazzolo, Saverio Paltrinieri, Piera Anna Martino, Vito Martella, Nicola Decaro

**Affiliations:** 1Laboratorio di Analisi Veterinarie MYLAV La Vallonea Passirana di Rho, Via Giuseppe Sirtori, 9, 20017 Rho, Italy; martamedardo@mylav.net (M.M.); walterbertazzolo@laboratoriolavallonea.it (W.B.); 2Department of Veterinary Medicine of Bari, University of Bari “Aldo Moro”, Strada per Casamassima Km 3, Valenzano, 70010 Bari, Italy; paolo.capozza@uniba.it (P.C.); vito.martella@uniba.it (V.M.); 3Department of Veterinary Medicine and Animal Sciences, University of Milan, 26900 Lodi, Italy; saverio.paltrinieri@unimi.it; 4One Health Unit, Department of Biomedical, Surgical and Dental Sciences, University of Milan, 20122 Milan, Italy; piera.martino@unimi.it

**Keywords:** septic exudate, diagnostic accuracy, cytology, culturomics, pets, MALDI-TOF

## Abstract

**Simple Summary:**

Septic exudates in the body cavities of dogs and cats are considered a critical clinical condition. The current elective diagnostic tools for detecting septic effusion are bacterial culture and fluid cytology. Although culture is considered the gold standard, clinicians may not have access to the results for several days. This may result in a delayed diagnosis of septic effusion, which may have adverse effects on patient outcomes. This study compared the performances of cytology and bacterial culture in the identification of septic exudative body cavity effusions in dogs and cats. The results of our investigation indicated moderate agreement between cytology and microbiology. Cytology and bacterial culture results for exudative body cavity effusions in dogs and cats can be misleading when conducted individually. To improve diagnostic accuracy, these two methodologies should be integrated.

**Abstract:**

The elective test for the determination of the effusions etiopathogenesis is represented by physico-chemical analysis and cytology. Nevertheless, the bacterial culture and antibiotic sensitivity tests are crucial for setting therapy and for the outcome. This study compared cytology with microbiology in the etiologic diagnosis of exudative body cavity effusions in dogs and cats collected from October 2018 to October 2022. All samples underwent aerobic and anaerobic culture and cytology examination. Bacterial identifications were confirmed using matrix-assisted laser desorption ionization-time of flight (MALDI-TOF) mass spectrometry, whereas cytological samples were blindly evaluated either in May Grunwald-Giemsa (MGG) or Gram-stained samples by two board-certified clinical pathologists. A moderate agreement (κ = 0.454) between cytology and bacterial culture was revealed. The sensitivity of the cytological evaluation in our study ranged from 38.5% to 67.9%, and the specificity ranged from 88.9% to 100%, depending on the type of the effusion, so cytology may not be representative of the etiopathogenesis, whereas bacterial culture can misidentify or fail to isolate the correct pathogen for difficult in vitro growing due to the presence of inhibitory substances or contamination. Cytology and bacterial culture results for exudative body cavity effusions in dogs and cats can be misleading if conducted separately, so these two tests should be performed together to increase diagnostic accuracy.

## 1. Introduction

Effusion represents a pathologic condition characterized by the accumulation of fluid in a body cavity (pleural, peritoneal, or pericardial) [1,2].

Patients are often presented to veterinarians with clinical signs secondary to fluid accumulation in the cavity, such as abdominal distension or dyspnea, clinical signs caused by the disease responsible for producing the effusion, or a combination of these conditions. Effusions may not be noticed by the pet owner until they become severe [2,3,4,5,6]. Therefore, the presence of an effusion usually is a clinical finding first seen by the practitioners during a physical examination. Clinical signs, such as dyspnea, lethargy, lack of stamina, and abdominal distension, should inspire the evaluation of the abdomen and thorax for an effusion [2,3,4,5,6].

Septic exudates in the body cavity in dogs and cats are considered a critical clinical condition for patients [2,3] and can originate from penetrating wounds, surgery, extension, the rupture of an adjacent infected lesion, or, infrequently, bacteremia. The most common causes of septic abdominal effusions are penetrating foreign bodies and the rupture of the gastrointestinal tract, with subsequent leakage of the intestinal content into the abdomen; this latter event can occur secondary to a penetrating foreign body or as a consequence of neoplasms, ulcers, or necrotic processes caused by tissue death following an intussusception [2,3]. On the other hand, for pleural effusions, the mechanisms of infection include lymphohematogenous spread during sepsis, extension from near anatomical sites (bronchopneumonia, parapneumonic spread, mediastinitis), or direct inoculation (migrating foreign body, thoracocentesis) [6]

Septic fluids are characterized by the presence of a high number of neutrophils and a smaller number of macrophages. Cell counts are typically greater than 13 × 10^9^ cell/L [1,2]. Most neutrophils are degenerated and/or toxic, possibly with evidence of bacterial phagocytosis, and free bacteria may be found in the background; however, bacteria that are not strong toxin producers (e.g., *Actinomyces* spp.) may be associated with non-degenerated neutrophils [4,7,8]. Cytology and bacterial culture represent the elective tests for the determination of the etiopathogenesis of effusions and their classification [2,3,9]. Cytological examination, however, cannot reliably identify bacterial species or predict their antibiotic sensitivity. Occasionally, specimens obtained from septic cavitary effusions contain high numbers of neutrophils without any bacteria being able to be microscopically identified. This may be due to recent or ongoing antibiotic treatment, the presence of a very low number of bacteria, or the presence of bacteria that are not visible because they are poorly stainable [10].

In cases in which cytology reveals the presence of bacteria or degenerated neutrophils, or when sepsis is suspected based on clinical history, a physical examination, or diagnostic imaging, despite the absence of bacteria or degenerated neutrophils in cytological specimens, it is imperative to submit a sample to the laboratory for analysis via culture test and antimicrobial susceptibility testing [11]. The most isolated bacterial species include aerobic or anaerobic bacteria, including *Clostridium* spp., *Bacteroides* spp., *Fusobacterium* spp., *Pasteurella* spp., and filamentous bacteria (*Nocardia* spp., *Actinomyces* spp.) [12]. Some of these bacteria are demanding and challenging to grow in cultures, so a negative culture result does not exclude the presence of a bacterial infection if the cytological findings are compatible with this possible diagnosis. When only extracellular bacteria are seen, and no degenerated neutrophils are present, it is possible that the sample has been contaminated during collection (pre-analytical phase) or that there is a hyperacute inflammatory process [12].

Thus, both cytology and culture may have important limitations in making an accurate diagnosis of a cavitary infection [2,3,9].

In addition to a cytological examination and bacteriological culture, several clinical biomarkers have been examined in plasma and effusions to enhance the prompt diagnosis of septic exudates [7]. Nevertheless, previous diagnostic markers have shown inconsistency in their ability to confirm septic effusion [7,13,14]. The comparison between glucose and lactate levels in peritoneal fluid and peripheral blood or plasma for diagnosing septic peritonitis has proven to be beneficial. However, recent studies have highlighted the limitations of these comparisons [7,8,15,16,17,18,19]. Alternative markers have been examined, but they have not been shown to be precise or accurate bedside testing [13,14,16,20,21,22,23,24,25,26,27].

This study aimed to assess the diagnostic accuracy of cytology, performed by two clinical pathologists employing two staining methods (May Grunwald-Giemsa MGG and Gram stain), in comparison to bacteriology, for the detection of bacterial infections in body fluid samples obtained from the pleural, peritoneal, or pericardial cavities of dogs and cats.

## 2. Materials and Methods

### 2.1. Sample Collection

Overall, in this retrospective study, 82 pleural, pericardial, and peritoneal effusions were collected from 25 dogs (14 pleural effusions, 2 pericardial effusions, and 9 peritoneal effusions) and 57 cats (9 peritoneal effusions and 48 pleural effusions) between October 2018 and October 2022. The collection of pleural, peritoneal, and pericardial effusions was conducted in private clinics by practitioners from client-owned patients whose health conditions require drainage, following the procedures recommended in the literature to avoid the contamination of the sample [2].

Effusion samples for cytology (approximately 1 mL) were collected in a tube with anticoagulant (k3-EDTA) and submitted to the laboratory, along with smeared slides made by veterinary practitioners at time of collection. A direct smear and a cytospin obtained through cytocentrifugation (Rotofix 32 A, Hettich, Tuttlingen, Germany) were made as soon as the specimens arrived in the laboratory.

Effusion samples for microbiology (approximately 1 mL) were collected in empty sterile tubes, in the syringe used for drainage, or in a swab with transport medium (Amies or Stuart, with or without charcoal). All samples were collected between 6 and 48 h before the arrival at the laboratory and were subsequently analyzed immediately after arrival.

The inclusion criteria for the sample were as follows: clinical suspicion of septic effusion, macroscopic findings potentially consistent with a septic exudate (e.g., clouded appearance, yellowish-brown coloration), and availability of data pertaining to species, age, sex, reproductive status, and breed. We excluded cases with missing data and all cases with clinical suspicion for non-septic effusion (e.g., uroperithoneum, transudates) in which cytology and microbiology were not requested on the same sample.

### 2.2. Effusion Evaluation and Cytological Analysis

For each sample, two slides were stained with the May Grunwald-Giemsa (MGG) stain (Bio-Optica Milano S.p.A., Milano, Italy), and two were stained with the Gram stain (Remel, Lenexa, KS, USA) following the procedures recommended in the manufacturer’s instruction. Cytological specimens were blindly evaluated by two board-certified clinical pathologists (W.B., S.P.), who classified either the Gram-stained or the MGG-stained slides as “positive” or “negative” for the presence of bacteria and grossly characterized the bacteria as cocci, rods, and filamentous (MGG stain) and as Gram-positive or -negative (Gram stain). The analysis of the cytological results was performed in two steps. First, the agreement between the two clinical pathologists in classifying the samples as positive or negative for bacteria was assessed on the whole caseload. Then, the agreement between cytology and bacteriology, which in the current study, was used as a gold standard, was assessed only on the samples for which there was complete agreement (positive for bacteria or negative for bacteria) between the two clinical pathologists.

### 2.3. Bacterial Culture

The bacterial culture results of exudate specimens were retrieved. The type of infection (either bacterial infection of single or mixed species) was recorded. Samples were cultured immediately without enrichment and also after enrichment at 37 °C in brain–heart infusion broth (BHI) (Oxoid, ThermoFisher Scientific, Monza, Italy). The BHIs were plated out after 24 h if they appeared turbid enough; otherwise, they were plated out after ten days of incubation. The methodology used for aerobic culture pre- and post-enrichment was the following: samples were cultured on Columbia sheep blood agar, Mac Conkey agar, Columbia nalidixic acid agar, Sabouraud dextrose agar, and plates were incubated for 24 to 48 h at 37 °C in aerobic conditions. For anaerobic culture samples, both pre- and post-enrichment samples were cultured on Schaedler agar (Oxoid, ThermoFisher Scientific, Italy), and plates were incubated for 24 to 48 h at 37 °C in anaerobic closed systems (Anaero Gen Compact-Thermo Fisher, Milan, Italy) that keep the atmosphere at 37 °C (98.6 F) and have less than 1% O_2_ and 8 to 14% CO_2_.

For microaerophilic culture samples, pre- and post-enrichment samples were cultured on Columbia sheep blood agar and incubated in a microaerophilic closed system (CampyGen Compact-Oxoid, ThermoFisher Scientific, Italy), providing an atmosphere of 8–10% carbon dioxide and 5–10% oxygen. All plates were incubated for seven days before reporting negative or no growth.

Bacterial isolates were initially classified according to colony morphologies, Gram staining, and microscopic appearances. MALDI-TOF mass spectrometry (VITEK^®^ MS-BioMèrièux, Florence, Italy) was subsequently utilized to validate the presumptive identifications. To assess each bacterium in quadruplicate, a spot of 1 microliter of sterile disposable loop was utilized to apply each colony to a VITEK^®^ MS DS plate. Subsequently, 1 microliter of VITEK^®^ MS CHCA matrix was added to each spot and allowed to dry. Following the application of the ATCC^®^ control *E. coli* strain as directed by the manufacturer, spectra were obtained from each sample by immediately reading the plate. With a reliability of up to 99.9%, the spectra represented by the bacterial protein pattern were digitized and automatically compared to the clinical spectra database (VITEK^®^ MS V3.2 Knowledge Base, Florence, Italy) to identify bacteria. When the results yielded a reliability of less than 96% or were inconsistent within quadruplicates, identifications were iterated until an acceptable score per quadruplicate was achieved.

To evaluate the overall accuracy of cytology, we used bacterial culture as the “gold standard” and defined all samples in which bacterial growth could be observed in at least one of the media as “positive” and all samples in which no isolates were found as “negative”.

The cytological test result was considered “positive” for all specimens in which both observers found phagocytosed and free bacteria (rods, cocci, or mixed types) on the MGG- and/or Gram-stained slides. Otherwise, cytology was considered “negative” in all samples in which both observers did not find phagocytosed and free bacteria.

Therefore, in the evaluation of the accuracy of cytology compared with the gold standard bacteriology, samples were considered “true positive” (TP) if cytology was positive in samples with positive bacteriology, “false positive” (FP) if cytology was positive in samples with negative bacteriology, “false negative” (FN) if cytology was negative in samples with positive bacteriology, and “true negative” (TN) if cytology was negative in samples with negative bacteriology.

### 2.4. Data Analysis

A statistical analysis of variables was performed using R software version 4.0.2 (R Foundation for Statistical Computing, Vienna, Austria; https://www.R-project.org/, accessed on 8 January 2024). Categorical data were summarized as counts and percentages. Continuous variables were summarized as mean and standard deviation when normally distributed and as median and range when non-normally distributed. The Kolmogorov–Smirnov and Shapiro–Wilk tests were used to verify the normal distribution of the continuous data.

Using the numbers of TP, FP, FN, TN, sensitivity (SN), and specificity (SP), negative predictive value (NPV), positive predictive value (PPV), diagnostic accuracy (DA), likelihood ratio of a positive test (LRP), and likelihood ratio of a negative test (LRN) were individually calculated for cytology. The agreement between the results of cytology and bacterial culture was evaluated with Cohen’s Kappa coefficient (k), which is interpreted as follows: values ≤ 0 indicating no agreement, values of 0.01–0.20 indicating none to slight, values 0.21–0.40 indicating fair, values 0.41–0.60 indicating moderate, values 0.61–0.80 indicating substantial, and values 0.81–1.00 indicating almost perfect agreement [28]. Moreover, Cohen’s Kappa coefficient (k) was calculated to evaluate the agreement between the two clinical pathologists’ evaluations.

## 3. Results

### 3.1. Signalment

The caseload included 25 dogs and 57 cats. Most of the dogs were purebred (56%; 14/25), female (56%; 14/25), and aged from 2 months to 13 years (median 5 years). Of these, 16% (4/25) were juvenile (≤12 months old), and 84% (21/25) were adult (>12 months old). However, most of the cats were domestic short-haired cats (84.2%, 48/57), male (61.4; 35/57), and aged from 2 months to 15 years (median 4 years). Of these, 15.8% (9/57) were juvenile, and 84.2% (48/57) were adult cats.

### 3.2. Cytological Analysis

Results recorded by the two observers in the MGG- or Gram-stained smears are reported in Table 1a,b. Using both stains, rods and mixed-type bacteria were the most common findings, while cocci alone were less common (Figure 1). Fungal hyphae, yeast cells, and parasites were not observed in any of the samples.

In MGG-stained specimens, bacteria were detected in 49 samples by one observer and in 50 by the other. In 28 samples, both observers did not find bacteria, in 45 cases, both the observers detected bacteria in cytological specimens, while in 9 cases, bacteria were detected by one of the two observers, but not by the other.

In Gram-stained specimens, bacteria were detected in 46 samples by one observer, and in 45 by the other. In 30 samples, both observers did not find bacteria, in 39 cases, both observers detected bacteria in cytological specimens, while in 13 cases, bacteria were detected by one of the two observers, but not by the other.

Considering both stains, bacteria were detected in MGG- and/or Gram-stained specimens in 50 cases by one observer and in 50 by the other. In 28 samples, both observers did not find bacteria, in 46 cases, both observers detected bacteria in cytological specimens, while in 8 cases, bacteria were detected by one of the two observers, but not by the other.

Overall, the concordance between the two observers was substantial, either for the MGG stain (κ = 0.770; 95% CI, 0.629–0.912) or the Gram stain (κ = 0.670; 95% CI, 0.519–0.839).

### 3.3. Bacterial Identification

Bacteria were isolated from 56.3% (44/82) of aerobic cultures and 29.2% (24/82) of anaerobic cultures for a total of 67 positive samples out of 82 (81.7%). In many positive samples (91%; 61/67), a single bacterial species was isolated, whereas multiple bacterial species were isolated from 9% (6/67) of positive samples; no bacteria were isolated in 18.3% (15/82) of samples. Molds and yeasts did not proliferate in any samples.

The most commonly aerobic bacteria isolated were *Pasteurella* spp. (11/44; 25%) and *E. coli* (8/44; 18.2%), followed by *Enterococcus* spp. (7/44; 15.9%), *Burkholderia* spp. (4/44; 9.1%), *Streptococcus* spp. (3/44; 6.8%), *Pseudomonas* spp. (3/44; 6.8%), *Klebsiella* spp. (2/44; 4.5%), *Staphylococcus* spp. (2/44; 4.5%), and one each of *Eggerthia* spp., *Moraxella* spp., *Stenotrophomonas* spp., *Enterobacter* spp., *Proteus* spp., *Serratia* spp., and unidentified Gram-positive and unidentified Gram-negative.

Anaerobic bacteria identified were *Bacteroides* (11/24; 45.8%), *Fusobacterium* spp. (10/24; 41.6%), *Clostridium* spp. (2/24; 8.3%), and one each of *Peptoniphilus* spp. and *Peptostreptococcus* spp.

Overall, 47 feline samples out of 57 were positive (82.4%). The most common bacterial isolate was *Bacteroides pyogenes* (9 *Bacteroides pyogenes* out of a total of 53 bacteria isolated; 16.98%), followed by *Pasteurella multocida* (7/53; 13.2%), *Fusobacterium russii* (6/53; 11.32%), and *E. coli* (3/53; 5.66%). With 2/53 (3.77%), there were *Enterococcus faecalis*, *Pseudomonas aeruginosa*, *Streptococcus pneumoniae*, *Fusobacterium nucleatum*, *Enterococcus faecium*, and *Bacteroides fragilis*. With only one isolation each (1/53; 1.88%), there were the following: *Klebsiella pneumoniae*, *Proteus mirabilis*, *Enterobacter cloacae*, unidentified Gram-negative, *Pasteurella canis*, *Clostridium bifermentans*, *Peptoniphilus asaccharolyticus*, *Staphylococcus pseudintermedius*, *Peptostreptococcus anaerobius*, *Streptococcus canis*, *Burkholderia cepacia*, *Burkholderia stabilis*, *Pseudomonas aeruginosa*, *Stenotrophomonas maltophilia*, unidentified Gram-positive, *Moraxella osloensis*, and *Serratia liquefaciens* (Table 2). Twelve Gram-positive bacteria and thirty-five Gram-negative bacteria resulted from feline thoracic effusions, and two Gram-positive and four Gram-negative bacteria resulted from abdominal effusions. All bacteria isolated from feline abdominal effusions were aerobe, while 23 bacteria isolated from feline thoracic effusions were anaerobe and 26 were aerobe.

Twenty canine samples out of twenty-five were positive (80%). The most common bacterial isolate was *E. coli* (5/20; 25%), followed by *Pasteurella canis* (3/20; 15%) and *Bacteroides pyogenes (2/20*; *10%).* With only one isolation each (1/20; 2%), there were the following: *Staphylococcus pseudintermedius*, *Burkholderia stabilis*, *Enterococcus faecium*, *Sphingomonas paucimobilis*, *Fusobacterium nucleatum*, *Fusobacterium russii*, *Eggerthia catenaformis*, *Klebsiella oxytoca*, *Burkholderia contaminans*, *Pasteurella multocida*, and *Enterococcus faecalis* (Table 3). Three Gram-positive bacteria and ten Gram-negative bacteria resulted from canine thoracic effusions, and one Gram-positive and six Gram-negative bacteria resulted from abdominal effusions. Only one bacterium isolated from canine abdominal effusions was anaerobe, and six were aerobe, while four bacteria isolated from canine thoracic effusions were anaerobe, and nine were aerobe.

In the two canine pericardial effusion samples, *Burkholderia stabilis*, a facultative anaerobic Gram-negative bacterium, was isolated in one case, while the other yielded a sterile result.

Considering the 74 samples with a complete agreement between the two observers (Table 4) in at least one staining, in the following 16 cases, bacteriological results and cytology were in disagreement: fifteen cases resulted positive in culture but negative for cytology, and only one case was positive according to the cytologic analysis, but with no significant growth in bacterial culture. The rate of agreement between cytology and bacteriology was slightly higher for slides stained with MGG than for slides stained with Gram.

Of the fifteen samples that tested positive on microbiology but not on cytology, it is interesting to note that eleven showed the growth of Gram-negative bacteria; of these, three were bacteria of the genus *Pasteurella* spp. while another three were anaerobic bacteria. The remaining four samples that tested positive on microbiology, but not on cytology, were detected as Gram-positive bacteria, as follows: two were bacteria belonging to the genus *Enterococcus* spp., one belonged to the genus *Staphylococcus* spp., and one was a Gram-positive bacterium for which was impossible to obtain a speciation.

### 3.4. Data Analysis

Overall, cytology itself had an SN of 75.0%, SP of 92.9%, PPV of 97.8%, NPV of 46.4%, DA of 78.4%, LRP of 10.5, and LRN of 0.27 compared to an ultimate diagnosis of septic versus non-septic exudates, based on bacteriology. The SN, SP, and DA of cytological examination for each type of effusion vary widely (Table 5).

The SN, SP, and DA of cytological examination for each type of effusion (peritoneal, pericardial, and pleural) considering the two different species are reported in Table 6 and Table 7.

Interestingly, the diagnostic performances of the cytological evaluation slightly differ between the two observers and between the different stains (Gram and MGG) (see Table 8), with Gram being less sensitive than MGG in detecting bacteria for both observers.

A moderate agreement between bacterial culture and cytology (Gram plus MGG) for determining septic exudates was established using the kappa statistic (κ = 0.454; 95% CI, 0.226–0.682) [28]. The comparison between microbiology and each single stain showed fair agreement with the Gram stain κ = 0.400 (95% CI, 0.171–0.629) and moderate agreement with MGG κ = 0.452 (95% CI, 0.223–0.680).

## 4. Discussion

Septic exudates in the body cavities of dogs and cats are common critical clinical conditions for patients [2,3,29]. Although cavitary effusions are routinely examined cytologically to evaluate the presence of inflammation and/or microorganisms, providing clinicians and owners with more rapid results, a bacterial culture is used to confirm the presence of infection and eventually identify the pathogen and assess its susceptibility profile in order to set the correct antimicrobial therapy [12]. In the current study, we evaluated the diagnostic accuracy of cytology as a tool for the diagnosis of septic exudates. Overall, using microbiology as the reference method, we found that cytology is poorly sensitive but quite specific for the detection of septic exudates, with some variations depending on the type of fluid likely due to its different cellular composition or the different types of bacteria detected in the different cavities [11]. Based on the high specificity and positive likelihood ratio, a positive cytology result can be considered confirmatory, while a negative cytological result does not exclude that bacteria are present but not seen by the observers. Therefore, cytology is more effective at confirming bacterial infection rather than excluding it.

Our findings confirm the results of a recent study that tested 244 fluids, such as blood, abdominal, thoracic, joint, and CSF, to evaluate the accuracy of cytology in detecting bacterial infections in body fluids. Indeed, cytology’s overall sensitivity was poor, but its specificity for sepsis was high, with differences in sensitivities and specificities calculated for each fluid type [11]. Allen et al. [11] found a very high specificity for abdominal fluid while finding that cytology’s sensitivity for the detection of bacteria causing bacterial infection in thoracic effusion was higher than the specificity. Indeed, the authors suggests that cytology is better at ruling out septic thorax than ruling it in, although the calculated sensitivity (87.5%) was moderate. In our evaluation of the cytological examination by two certified clinical pathologists, differences in sensitivity and SN, SP, NPV, PPV, DA, LRP, and LRN were found in the evaluation of smears stained with MGG and Gram stain. This aspect remains a major limitation of cytological examinations in that the accuracy of the diagnosis depends mainly on the skill and experience of the observer [12,30]. However, substantial concordance was found between the cytological evaluations of the two certified clinical pathologists, giving us evidence that interobserver variability is not a major limitation in routine practice.

Overall, we found a greater sensitivity in smears stained with MGG compared with those stained with Gram stain, while no differences were detected for specificity. Likewise, the diagnostic accuracy appears to be greater for smears stained with MGG than for those stained with Gram. That is reasonable since MGG stains all bacteria with a clear blue/purple color that is easier to detect, while the alternative stain dyes Gram-negative bacteria with a pale red color that can be difficult to evaluate. In our study, we focused on the diagnosis of sepsis/non-sepsis of an exudate, comparing cytological results with microbiological results as follows: the preliminary observations on the use of Gram staining certainly need to be characterized more specifically with further comparative studies between morphology and bacterial identification resulting in microbiology

De Brauwer et al. [31] showed that the greatest interobserver concordance was achieved with MGG staining alone, which is close to the results obtained in our study. The real advantage of performing both staining options in the case of septic effusion is being able to assess in advance the bacterial type involved in the infection; this could be useful for the purpose of possible therapy pending microbiological results and is essential to attribute greater significance to microbiological isolation, possibly discriminating a contaminated growth.

This study presents some limitations that may have affected the results. First, the medical records were not always available, and this study was conducted retrospectively. Another limitation was the strict reliance on culture as the reference method to identify bacterial organisms. Bacterial culture can be challenging and may require several days, rendering it impracticable in some cases for timely clinical decision-making regarding surgical requirements. Indeed, it should be noted that certain instances of confirmed septic effusion may produce negative cultural outcomes [29,32,33,34,35,36]; failure to isolate the correct pathogen for difficult in vitro growth in the presence of inhibitory substances (mostly antibiotic molecules) or polymicrobial infections is a frequent occurrence. Also, purulent exudate charged with neutrophilic lytic enzymes has an inhibitory effect on bacterial in vitro growth. For our case collection, we are not aware of how many patients were actually on antibiotic therapy or not, resulting in an additional limitation that, however, mimics what routinely happens in the caseload of a diagnostic veterinary laboratory. On the other hand, negative cytology does not always rule out the presence of septic exudate because the visualization of bacteria under a light microscope is subject to the bacterial load present in the specimen, which below certain thresholds, is undetectable, especially with abundant inflammatory cells in the background or in the presence of scant material. Another limitation was the variability in sending samples; some of them were taken in swabs with a transport medium, and others were taken in sterile syringes or sterile empty tubes. These differences may have affected the bacterial viability and their growth in vitro.

Interestingly, in our investigation, the most commonly isolated strains in feline samples were *Pasteurella* spp. for aerobic isolates and *Bacteroides* spp. for anaerobic isolates, while *Escherichia coli* was the most commonly isolated strain for canine aerobe and *Bacteroides* spp. was the most commonly isolated strain for anaerobe isolates. These findings, apparently discrepant, may reflect the different proportions in our study between thoracic samples (prevalent in the feline species) and abdominal samples (prevalent in the canine species). In particular, the high prevalence of feline pyothorax may have biased the results regarding cats. However, in general, these trends are in agreement with the literature [37,38] *Pasteurella* spp. and *Bacteroides* spp. were reported as the most commonly isolated strains from both canine and feline septic effusions.

Polymicrobial infections, sustained by both aerobic and anaerobic bacteria, appear to drive the majority of pleural effusions [12]. The population of isolated organisms may also be affected by geographical location [39,40]. Moreover, mycoplasma species may potentially cause pneumonia and pyothorax in kittens and immunosuppressed adults [2,3,12]. Like in septic pleural effusions, in most septic peritonitis cases, multiple microorganisms are often isolated, although Gram bacteria are predominant [12].

Given the scarcity of pericardial infections, it is crucial to acknowledge that drawing significant conclusions is not possible due to the inadequate number of isolates.

The moderate level of agreement between cytology and microbiology reiterates the importance of combining the two examinations in the context of a diagnosis of septic effusion. Most of the samples that were cytologically negative showed the growth of *Pasteurella* spp. or anaerobe Gram-negative bacteria that are either very small in size or filiform; it could be difficult to see them, especially if they are in small numbers or in the presence of abundant inflammatory cells, which is very common in septic effusions.

Several opportunistic pathogens have been isolated, some of them with low known pathogenicity and capable of inducing infection only in immunocompromised patients (e.g., *Sphingomonas paucimobilis*, *Moraxella osloensis*, *Burkholderia cepacia*), and all have rarely been identified in septic exudates. In four out of seven cases in which an opportunistic bacterium was isolated, there was a disagreement between negative cytology and positive microbiology. We cannot exclude the possibility of post-sampling contamination, which is easier when there is time between collection and processing since these are mostly environmental organisms [41,42,43]; alternatively, the bacterial load in the sample was so small that it could not be identified under the microscope, but with the enrichments carried out in microbiology isolation, identification was nevertheless achieved.

The results of our investigation indicated a moderate agreement between cytology and microbiology. It is possible that cytology does not provide an accurate picture of the pathogenesis because of a low bacterial load in the sample or the presence of small/filiform bacteria that are difficult to evaluate. A bacterial culture may misidentify or fail to isolate the correct pathogen due to contamination, inhibitory substances, or in vitro fastidious growth conditions. Our study started with the assumption that bacteriology was the gold standard for the diagnosis of septic effusions; in light of the results obtained, it was found that both bacteriology and cytology have limitations, especially when taken individually, and it is appropriate to perform the two examinations combined together in presence of septic effusions to increase the diagnostic chances.

## 5. Conclusions

In conclusion, the diagnosis of septic exudate remains a challenge that involves various specialists who have to work in sync to reach an etiological diagnosis in the shortest possible time to guarantee patients the best care as soon as possible. The outcomes of this retrospective analysis of cytology and microbiology accuracy are anticipated to provide valuable insights for subsequent research endeavors focused on the development of novel diagnostic assays to promptly identify bacterial sepsis exudates in feline and canine patients. Research could prospectively assess the accuracy of cytology as a method for identifying bacterial organisms, thereby circumventing a number of these constraints.

## Figures and Tables

**Figure 1 animals-14-01762-f001:**
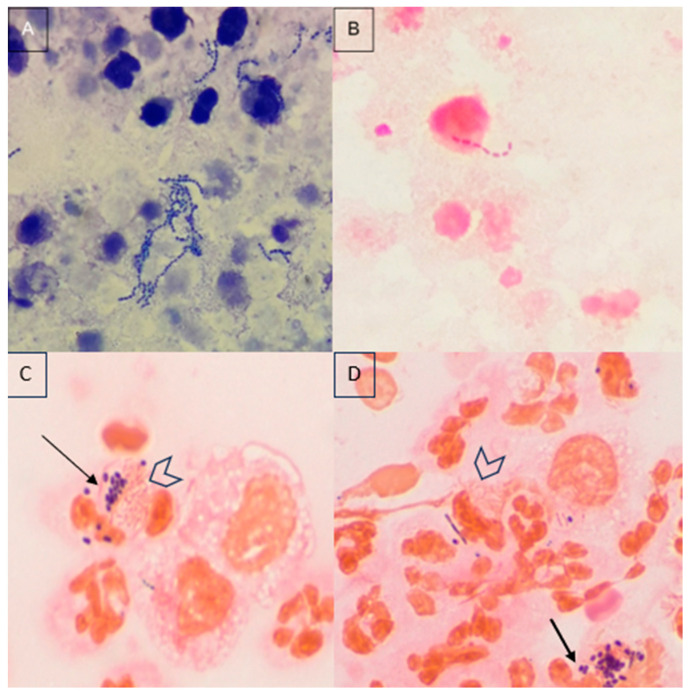
Examples of positive cytological results. Panel (**A**): free and phagocytized cocci arranged in chains with degenerated neutrophils. Pleural effusion, dog. May Grunwald-Giemsa stain, 100x. Panel (**B**): chain of Gram-negative rods inside a degenerated neutrophil. Peritoneal effusion, dog. Gram stain, 100x. Panel (**C**,**D**): mixed clusters of Gram-positive cocci (arrows) and Gram-negative rods (arrowheads) inside neutrophils. Pleural effusion, cat. Gram stain, 100x.

**Table 1 animals-14-01762-t001:** Results regarding the concordance of the cytological evaluation of the two observers in the 82 May Grunwald-Giemsa (MGG)-stained slides (a). Results regarding the concordance of the cytological evaluation of the two observers in the 82 Gram-stained slides (b).

(a)		Observer 1			
		Cocci	Rods	Mixed	Negative
**Observer 2**	**Cocci**	2	1	1	0
	**Rods**	3	10	6	1
	**Mixed**	1	5	16	3
	**Negative**	1	2	2	28
**(b)**		**Observer 1**			
		**Cocci**	**Rods**	**Mixed**	**Negative**
**Observer 2**	**Cocci**	3	1	2	1
	**Rods**	1	10	6	2
	**Mixed**	2	2	12	4
	**Negative**	3	2	1	30

**Table 2 animals-14-01762-t002:** Bacteria isolated from 47 feline positive body cavity effusions. A: aerobe; AN: anaerobe.

Bacteria Isolated	Number of Cases	Percentage	Gram/A/AN
*Bacteroides pyogenes*	9	16.98	Gram−, AN
*Pasteurella multocida*	7	13.2	Gram−, A
*Fusobacterium russii*	6	11.32	Gram−, AN
*Escherichia coli*	3	5.66	Gram−, A
*Enterococcus faecalis*	2	3.77	Gram+, A
*Enterococcus faecium*	2	3.77	Gram+, A
*Pseudomonas aeruginosa*	2	3.77	Gram−, A
*Streptococcus pneumoniae*	2	3.77	Gram+, A
*Fusobacterium nucleatum*	2	3.77	Gram−, AN
*Bacteroides fragilis*	2	3.77	Gram−, AN
*Klebsiella pneumoniae*	1	1.88	Gram−, A
*Proteus mirabilis*	1	1.88	Gram−, A
*Enterobacter cloacae*	1	1.88	Gram−, A
*Pasteurella canis*	1	1.88	Gram−, A
*Clostridium bifermentans*	1	1.88	Gram+, AN
*Peptoniphilus* *asaccharolyticus*	1	1.88	Gram+, AN
*Staphylococcus* *pseudintermedius*	1	1.88	Gram+, A
*Peptostreptococcus* *anaerobius*	1	1.88	Gram+, AN
*Streptococcus canis*	1	1.88	Gram+, A
*Burkholderia cepacia*	1	1.88	Gram−, A
*Burkholderia stabilis*	1	1.88	Gram−, A
*Serratia liquefaciens*	1	1.88	Gram−, A
*Stenotrophomonas maltophilia*	1	1.88	Gram−, A
*Moraxella osloensis*	1	1.88	Gram−, A
Unidentified Gram positive	1	1.88	Gram+, A
Unidentified Gram negative	1	1.88	Gram−, A

**Table 3 animals-14-01762-t003:** Bacteria isolated from 20 canine-positive body cavity effusions. A: aerobe; AN: anaerobe.

Bacteria Isolated	Number of Cases	Percentage	Gram/A/AN
*Escherichia coli*	5	23.80	Gram−, A
*Pasteurella canis*	3	14.28	Gram−, A
*Bacteroides pyogenes*	2	9.52	Gram−, AN
*Pasteurella multocida*	1	4.76	Gram−, A
*Staphylococcus* *pseudintermedius*	1	4.76	Gram+, A
*Enterococcus faecalis*	1	4.76	Gram+, A
*Burkholderia contaminans*	1	4.76	Gram−, A
*Burkholderia stabilis*	1	4.76	Gram−, A
*Eggerthia catenaformis*	1	4.76	Gram+, AN
*Fusobacterium russii*	1	4.76	Gram−, AN
*Fusobacterium nucleatum*	1	4.76	Gram−, AN
*Klebsiella oxytoca*	1	4.76	Gram−, A
*Sphingomonas paucimobilis*	1	4.76	Gram−, A
*Enterococcus faecium*	1	4.76	Gram+, A

**Table 4 animals-14-01762-t004:** Agreement between bacteriology and cytology.

Cytology		Bacteriology	
		Positive (%)	Negative (%)
**MGG (*n* = 73)**	**Positive**	44 (60.3%)	1 (1.4%)
	**Negative**	15 (20.5%)	13 (17.8%)
**Gram (*n* = 69)**	**Positive**	38 (55.1)	1 (1.5%)
	**Negative**	17 (24.6%)	13 (18.8%)
**MGG or Gram (*n* = 74)**	**Positive**	45 (60.8%)	1 (1.4%)
	**Negative**	15 (33.6)	13 (17.6%)

**Table 5 animals-14-01762-t005:** Diagnostic performances of the cytological evaluation in each type of effusion on the whole caseload.

Test Parameters	Pleural (n = 57)	Peritoneal (n = 15)	Pericardial (n = 2)
Sensitivity	79.2%	63.6%	0.0%
Specificity	88.9%	100.0%	ND
Positive predictive value	97.4%	100.0%	ND
Negative predictive value	44.4%	50.0%	50.0%
Diagnostic accuracy	80.7%	73.3%	50.0%
Likelihood ratio of a positive test	7.13	Inf	ND
Likelihood ratio of a negative test	0.23	0.36	ND

ND: not determinable; Inf: infinite; n: number of positive.

**Table 6 animals-14-01762-t006:** Diagnostic performances of the cytological evaluation in each type of effusion in canine samples.

Test Parameters	Pleural (n = 12)	Peritoneal (n = 7)	Pericardial (n = 2)
Sensitivity	70.0%	80.0%	0.0%
Specificity	100.0%	100.0%	ND
Positive predictive value	100.0%	100.0%	ND
Negative predictive value	40.0%	66.7%	50.0%
Diagnostic accuracy	75.0%	85.7%	50.0%
Likelihood ratio of a positive test	Inf	Inf	ND
Likelihood ratio of a negative test	0.30	0.20	ND

ND: not determinable; Inf: infinite; n: number of positive.

**Table 7 animals-14-01762-t007:** Diagnostic performances of the cytological evaluation of each type of effusion in feline samples.

Test Parameters	Pleural (n = 45)	Peritoneal (n = 8)
Sensitivity	81.6%	50.0%
Specificity	85.7%	100.0%
Positive predictive value	96.9%	100.0%
Negative predictive value	46.2%	40.0%
Diagnostic accuracy	82.2%	62.5%
Likelihood ratio of a positive test	5.71	Inf
Likelihood ratio of a negative test	0.21	0.50

ND: not determinable; Inf: infinite; n: number of positive.

**Table 8 animals-14-01762-t008:** Diagnostic performances of the cytological evaluation for the two observers and each type of stain.

Test Parameters	Observer 1		Observer 2	
	Gram (n = 46)	MGG (n = 49)	Gram (n = 45)	MGG (n = 50)
Sensitivity	64.2%	71.6%	65.7%	70.1%
Specificity	86.7%	86.7%	86.7%	86.7%
Positive predictive value	95.6%	96%	95.7%	95.9%
Negative predictive value	35.1%	40.6%	36.1%	39.4%
Diagnostic accuracy	68.3%	74.4%	69.5%	73.2%
Likelihood ratio of a positive test	4.81	5.37	4.92	5.27
Likelihood ratio of a negative test	0.41	0.32	0.39	0.39

n: number of positive.

## Data Availability

Raw data are available upon request.

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
