# Peer review of "Diagnosis of Septic Body Cavity Effusion in Dogs and Cats: Cytology vs. Bacterial Culture"

_animals, 2024, doi:10.3390/ani14121762_

Round 1

Reviewer 1 Report

Comments and Suggestions for Authors

Thank you for this submission - I can tell the authors put a lot of work into this manuscript, but some factors require clarification in order for the reader to make meaningful conclusions from your data. Overall, my 2 main comments (with specifics listed below) are: 1. the study design/methodology is very unclear, and does not clearly relate to the aim of the study; 2. It is unclear to me how this study is novel as compared to Allen et al. 2022, which is listed as reference 11 in the manuscript. 

- Lines 49-51: while I agree that this statement in my experience is true, it should either be cited with literature reference(s) or have the wording changed to be less declarative (such as "Effusions may not be noticed by the pet owner.." etc.)

- Lines 58-59: also foreign body perforations as a common cause of septic peritoneal effusion

- Line 81: I think "antimicrobial susceptibility testing" for AST is the more correct term, not "antibiotic sensitivity testing." Also, you should check with the guidelines of the journal, but if AST is not frequently referenced, I would not abbreviate it. This part of the intro gives the reader the idea that you're going to say something about AST, but the methods clearly are comparing culture results to cytology results, and AST is irrelevant to your study (or not included). 

- Line 81-83: while I would agree with this statement in the case of pyothorax, I do not believe that is true in abdominal sepsis. The sentence as stated requires citations. - see other comment below regarding the terminology for "facultative anaerobe" vs. "aerobe."

- Line 98: I'm not sure the cytology itself is "the cornerstone" of diagnosis, but rather cytology plus culture, as your study suggests. I would suggest that the sentence be changed to a statement which is less declarative - something like "Thus, both cytology and culture may have important limitations in making an accurate diagnosis of cavitary infection," and then I would include in the introduction the precise limitations of both of these tests to set up the reader for why you looked at these cases retrospectively in the first place.

- Line 100-102: I don't think it's appropriate to have a study aim "to provide clear advice for the correct diagnosis" in a retrospective study design. This warrants some editing. You could state that your aim was to assess the level of agreement between cytology and culture in patients with definitive evidence of cavitary infection, and depending on the result, you can recommend prospective study based on your retrospective assessment, but your study aim cannot be a diagnostic recommendation given the study design. 

- Line 128: given the study design is retrospective, "adequate volume of effusion" is not a relevant inclusion criteria (if fluid was submitted for the tests you described above, it was reviewed).

- What about exclusion criteria? Did you include all samples that either had cytology or culture or both supporting an infectious etiology? This needs to be better described.

- Line 140/141: I don't understand why disagreement between pathologists led to exclusion of the case. It would likely help for you to describe your "case definition" of what you are calling a septic vs. non-septic effusion. This paragraph alludes to using culture as the gold standard, comparing cytology findings with that? For example, if you had a culture sample from the abdomen from a case of a definitively diagnosed intestinal perforation which grew E. coli and Enterococcus, but the pathologists disagreed in their cytology in that one declared "gram negative rods" and the other "gram positive cocci" noted, they would both likely be correct, but per your statement here, the case would have been excluded? Similarly, if you only included cases where the pathologists agreed, it doesn't make sense to analyze agreement between the 2 clinical pathologists, as they already agree by definition. Often these cavitary infections are polymicrobial. So I think there is a lot about the materials and methods that needs to be described more clearly/thoroughly.

- Line 144: I would eliminate "Antimicrobial Susceptibility" from the title, as you did not report AST in your results, just speciation/culture results (this is perfectly fine, but the focus of your study is on the culture/bacterial isolation, not the AST). 

- Line 147: is this the methodology just for anaerobic cultures specifically? I would make that clear - something like "The methodology used for anaerobic cultures was..." and in the following paragraph, similarly introduce that that methodology was for aerobic culture.

- Lines 177-182: I would argue this section belongs in section 2.3, not the stats. However, the description in these lines is unclear how you defined certain findings. I recommend clearly stating what test/result you were using as your "gold standard" to compare to (knowing that it is unlikely a true gold standard) and then specific definitions of "positive cytology test" and "true positive cytology test." It's unclear as written right now what the difference is or why it matters.

- Line 196: it's most acceptable to report descriptive statistics such as mean vs. median years accompanied by standard deviation or range respectively. You also never mentioned in your stats section how you were determining if the data for each variable such as these were normally distributed (Shaprio Wilk test? Something similar?). You should also include a statement in your stats section along the lines of "continuous patient data are reported as mean +/- standard deviation when normally distributed and as median (range) when non-normally distributed" or something to that effect. But reporting both the mean and median is not customary.

- Table 1: again, a test of normality needs to be performed and reported for these continuous data, and the appropriate descriptive statistic reported. If that was performed and all were normally distributed, report the mean and SD. If non-normally distributed, report the median and range. Another question I have is whether you hypothesize that something about these variables (SG, TNCC, RBC, Ht, TP, LDH) affected the interpretation of cytology as septic vs. non-septic. Otherwise, why report them at all? It would make more sense to me to have certain cutoffs for the fluid analysis as deeming it non-septic (very low TNCC, low LDH, etc.). But as of now, it's unclear why you're reporting the results of the fluid analysis itself when what you aim to compare is the pathologist's cytology results vs. the culture result.

- Lines 208-213: Why are the results of the gram stain not incorporated into this? Specifically if rods were most common, what is the breakdown of gram positive vs. gram negative rods - those have very different implications when you're comparing to the results of cultures (as an example, if the pathologist read out of cytology says "gram positive rods" and the culture result is E. coli, the pathologist was wrong and those results are discordant, and you should comment on why that might be in the discussion).

- Table 2: it's unclear to me what "Observer 2" indicates when the title of the table implies that these are results from both observers? Are these just the results they agreed upon?

- Figure 1: as you alluded to in your intro, one cannot claim that figure 1 panel A is indeed streptococci, only that it was "gram positive cocci arranged in chains." If the culture result from that same fluid cytology agrees that it is Streptococcus, that's great, but you cannot claim that speciation from the picture alone. 

- Also regarding figure 1/your methods: if bacteria were noted specifically only intracellularly, how did you interpret the cytologic conclusion of a Gram stain? in theory, a Gram stain can only help delineate between gram positive vs. gram negative bacteria when they are extracellular. So for figure 1 panel B for example, how can you conclude that that was gram negative rods, when it should not pick up the Gram stain if it's intracellular bacteria regardless of whether it's truly a gram positive or gram negative rod bacterial species?

- Line 224/225: This sentence as written does not make sense. What are you trying to relay here?

- Table 3/general comment: while some of the organisms you listed such as E. coli may be categorized as a facultative anaerobe, some of the confusion in your introduction and your categorization here may be because most clinicians would consider an organism like E. coli to be an aerobic organism (yes, facultative anaerobe is a subtype of this, but given that you didn't further categorize organisms into microaerophiles, obligate aerobes, aerotolerant anaerobes etc., I would recommend amending your categorizations to strictly "anaerobes" and "aerobes" to limit this confusion. This may vary by geographic location as well, but where I live and practice, if a trainee claimed that an organism like E. coli was "an anaerobe" when what they meant was "facultative anaerobe", I would pronounce them incorrect.

- Table 5: in all 3 of these comparisons (MGG, gram, either), there are relatively high false negative rates (20.5%, 24.6%, 33.6%) - were these mostly in anaerobic infections? We talk about how anaerobic cultures in theory have higher false negative rates, but in this table, the message I am getting is that there were a good number of samples which were negative for bacteria on cytology but had positive growth - I would recommend commenting on that subpopulation of false negative cytologies and the types of organisms they grew out on culture testing to look for any patterns or commonalities. Conversely, were there cases where both pathologists agreed that bacteria were present in the sample, but the culture result was negative?

- Table 6: given the very few pericardial infections, it should be noted in the discussion that no meaningful conclusions can be made due to too few isolates.

- General comment: in addition, since it seems that the majority of your study population was feline pyothorax, that should be discussed as a limitation in your discussion.

- Line 301-302: Looking at the parameters reported in Table 9, I do not think this statement is correct (the 2 observers seem extremely close in all parameters when you look at gram vs. MGG stains separately - for example, Sn of gram stains was 64.2% observer 1 and 65.7% observer 2; PPV for MGG was 96% for observer 1, 95.9% for observer 2, etc.). Again in your discussion on Lines 326-332 you claim this, which I believe is not correct.

- Line 316: culture is also used to confirm or refute the presence of infection, I would argue, not just to guide antimicrobial therapy.

- I had the impression from your methods description that you were going to assess whether the shape/gram stain criteria on cytology matched the organism identified on culture (e.g. were their gram positive cocci on cytology which was confirmed as Staphylococcus on culture [concordant result] vs. confirmed as E. coli on culture [discordant result]). I don't see this type of analysis described in the results - if that was not the intention, please clarify the methodology as mentioned in comments above. 

- Lines 367-371: Don't you think that the most common isolates for cats were due to the large proportion of pyothorax cases? We do expect typically a different distribution of the frequencies of certain bacterial species depending on the location and thus the source of infection. Many septic effusion studies in dogs have focused on abdominal sepsis, as those cases often are associated with a worse prognosis and often more severe disease compared with canine pyothorax. I suggest commenting on this discrepancy in this area of the discussion.

- Line 394: you focused on culture being the reference diagnostic and comparing cytology results to it, but did not comment much on the limitations of either test alone to substantiate this comment: "...in light of the results obtained it was found that both bacteriology and cytology have limitations especially when taken individually.." etc.

- You have not compared your study design and findings to that of Allen et al. 2022, which I think is warranted.

Comments on the Quality of English Language

While the text is relatively easy to follow, I do recommend a native English speaker review the manuscript after adjustments to the above comments/concerns are made to increase the clarity of the content.

Author Response

Dear Editor,

thank you for giving us the opportunity to submit a revised draft of our manuscript titled: “Diagnosis of Septic Body Cavity Effusion in Dogs and cats: Cytology Vs Bacterial Culture”, Manuscript ID: animals-2982837, submitted to Animals. We appreciate the time and effort that you and the reviewers have dedicated to providing your valuable feedback on our manuscript. We are grateful to the reviewers for their insightful comments on our paper. We have been able to incorporate changes to reflect most of the suggestions provided by the reviewers. We have highlighted the changes within the manuscript.

Here is a point-by-point response to the reviewers’ comments and concerns.

Comments from Reviewer 1 (R.1)

R.1.1: Thank you for this submission - I can tell the authors put a lot of work into this manuscript, but some factors require clarification in order for the reader to make meaningful conclusions from your data. Overall, my 2 main comments (with specifics listed below) are: 1. the study design/methodology is very unclear, and does not clearly relate to the aim of the study.

Reply to R.1.1: Thank you for your thoughtful review of our manuscript. We greatly appreciate your recognition of the effort invested in our research. We aimed to compare two diagnostic methodologies to enhance the understanding of how to implement the diagnosis of septic body cavity effusions in dogs and cats.

Currently diagnosis of septic body cavity effusion in dogs and cats typically involves a combination of cytology and bacterial culture. Cytology involves examining a sample of the effusion fluid under a microscope to identify the types of cells present. In cases of septic effusion, the presence of certain types of cells, such as neutrophils (a type of white blood cell), can indicate inflammation and possibly infection. Additionally, the presence of bacteria within the fluid or associated with the cells can suggest a bacterial infection. Bacterial culture involves taking a sample of the effusion fluid and attempting to grow bacteria from it in a laboratory setting. This can help identify the specific bacterial species causing the infection. Culturing the bacteria also allows for testing of antibiotic susceptibility, which can drive treatment decisions. Each method has its advantages and limitations. Cytology is a relatively quick and inexpensive method that can provide rapid preliminary information about the nature of the effusion. While it may not always detect low levels of bacteria, and it cannot provide information about the specific bacterial species causing the infection. Bacterial culture provides definitive identification of the bacterial species causing the infection and allows for antibiotic susceptibility testing. However, it can take several days to obtain results, which may delay treatment initiation. Additionally, culture may be less successful if the sample is contaminated or if the bacteria present are difficult to culture. In clinical practice, both cytology and bacterial culture are often used together to maximize diagnostic accuracy. Cytology can provide immediate information to guide initial treatment decisions, while bacterial culture provides more detailed information for targeted treatment and management of the infection.

We acknowledge your concern regarding the clarity of our study design and methodology. Our primary objective was to assess the effectiveness of these diagnostic approaches in identifying and managing septic effusions in veterinary patients. We understand the importance of providing clearer explanations and will work to ensure that our methodology is more closely aligned with the aims of the study.

We value your feedback and will make every effort to address your concerns and improve the overall clarity and coherence of our manuscript.

R.1.2: It is unclear to me how this study is novel as compared to Allen et al. 2022, which is listed as reference 11 in the manuscript. 

Reply to R1.2: Thank you for your insightful comment. While we greatly appreciate the work of Allen et al. 2022, which served as inspiration for our study, we aimed to contribute further insights by comparing the results of bacterial culture and cytology, assessed by two different observers and using two different staining techniques.

As you rightly pointed out, the diagnostic accuracy of these tests depends significantly on the observer's skills and experience. Additionally, the MGG staining technique is not routinely performed in clinical practice. Therefore, our study sought to evaluate the cytological assessment of slides stained with both Gram and MGG stains to explore the potential implementation of diagnosing septic effusions in dogs and cats.

We believe that our approach provides valuable insights into the diagnostic process and may offer practical implications for veterinary clinicians.

Once again, we sincerely appreciate your feedback and will make every effort to address your concerns effectively in our manuscript revisions.

R.1.3: - Lines 49-51: while I agree that this statement in my experience is true, it should either be cited with literature reference(s) or have the wording changed to be less declarative (such as "Effusions may not be noticed by the pet owner." etc.)

Reply to R1.3: We agree with the comments of reviewer and have checked and corrected the revised version of the manuscript. Furthermore, the references referred to in this sentence are reported at the end of the paragraph and are: 1. O'Brien, P.J.; Lumsden, J.H. The cytological examination of body cavity fluids. Semin Vet Med Surg Small Anim 1988, 3, 140-156. 2. Dempsey, S.M.; Ewing, P.J. A review of the pathophysiology, classification, and analysis of canine and feline cavitary effusions. J Am Anim Hosp Assoc 2011, 47, 1-11, doi:10.5326/JAAHA-MS-5558. 3. Stillion, J.R.; Letendre, J.A. A clinical review of the pathophysiology, diagnosis, and treatment of pyothorax in dogs and cats. J Vet Emerg Crit Care (San Antonio) 2015, 25, 113-129, doi:10.1111/vec.12274. 4. Alleman, A.R. Abdominal, thoracic, and pericardial effusions. Vet Clin North Am Small Anim Pract 2003, 33, 89-118, doi:10.1016/s0195-5616(02)00057-8. 5. Barrs, V.R.; Beatty, J.A. Feline pyothorax - new insights into an old problem: part 1. Aetiopathogenesis and diagnostic investigation. Vet J 2009, 179, 163-170, doi:10.1016/j.tvjl.2008.03.011. 6. Barrs, V.R.; Allan, G.S.; Martin, P.; Beatty, J.A.; Malik, R. Feline pyothorax: a retrospective study of 27 cases in Australia. J Feline Med Surg 2005, 7, 211-222, doi:10.1016/j.jfms.2004.12.004.

R.1.4: - Lines 58-59: also foreign body perforations as a common cause of septic peritoneal effusion

Reply to R1.4: We agree with the comments of reviewer and have modified the revised version of the manuscript.

R.1.5: - Line 81: I think "antimicrobial susceptibility testing" for AST is the more correct term, not "antibiotic sensitivity testing." Also, you should check with the guidelines of the journal, but if AST is not frequently referenced, I would not abbreviate it. This part of the intro gives the reader the idea that you're going to say something about AST, but the methods clearly are comparing culture results to cytology results, and AST is irrelevant to your study (or not included). 

Reply to R1.5: We agree with the comments of reviewer and have modified the revised version of the manuscript.

R.1.6: - Line 81-83: while I would agree with this statement in the case of pyothorax, I do not believe that is true in abdominal sepsis. The sentence as stated requires citations. - see other comment below regarding the terminology for "facultative anaerobe" vs. "aerobe."

Reply to R1.6: We agree with the comments of reviewer and have modified the revised version of the manuscript.

R.1.7: - Line 98: I'm not sure the cytology itself is "the cornerstone" of diagnosis, but rather cytology plus culture, as your study suggests. I would suggest that the sentence be changed to a statement which is less declarative - something like "Thus, both cytology and culture may have important limitations in making an accurate diagnosis of cavitary infection," and then I would include in the introduction the precise limitations of both of these tests to set up the reader for why you looked at these cases retrospectively in the first place.

Reply to R1.7: We would like to point out to the reviewer that at lines 71-72: “Cytological examination however cannot reliably identify bacterial species or predict their antibiotic sensitivity.” and lines 84-89: “Some of these bacteria are demanding and challenging to grow in culture, so a negative culture result does not exclude the presence of a bacterial infection if the cytological findings are compatible with this possible diagnosis. When only extracellular bacteria are seen and no degenerated neutrophils are present, it is possible that the sample has been contaminated during collection (pre-analytical phase) or that there is a hyperacute inflammatory process” we have indicated the respective limitations of the two tests which would justify our comparison. But we understand the revisor’s point of view and to avoid misunderstanding we have modified the revised manuscript.

R.1.8: - Line 100-102: I don't think it's appropriate to have a study aim "to provide clear advice for the correct diagnosis" in a retrospective study design. This warrants some editing. You could state that your aim was to assess the level of agreement between cytology and culture in patients with definitive evidence of cavitary infection, and depending on the result, you can recommend prospective study based on your retrospective assessment, but your study aim cannot be a diagnostic recommendation given the study design. 

Reply to R1.8: According to the reviewer's kind suggestions and comments, we have modified the revised manuscript.

R.1.9: - Line 128: given the study design is retrospective, "adequate volume of effusion" is not a relevant inclusion criteria (if fluid was submitted for the tests you described above, it was reviewed).

Reply to R1.9: We agree with the comments of reviewer and have modified the revised version of the manuscript.

R.1.10: - What about exclusion criteria? Did you include all samples that either had cytology or culture or both supporting an infectious etiology? This needs to be better described.

Reply to R1.10: According to the reviewer's kind suggestions and comments, we have specified the “inclusion and exclusion criteria used” at lines 133-136 “We excluded cases with missing data and all cases with clinical suspicion for non-septic effusion (e.g. uroperithoneum, transudates).”

R.1.11: - Line 140/141: I don't understand why disagreement between pathologists led to exclusion of the case. It would likely help for you to describe your "case definition" of what you are calling a septic vs. non-septic effusion. This paragraph alludes to using culture as the gold standard, comparing cytology findings with that? For example, if you had a culture sample from the abdomen from a case of a definitively diagnosed intestinal perforation which grew E. coli and Enterococcus, but the pathologists disagreed in their cytology in that one declared "gram negative rods" and the other "gram positive cocci" noted, they would both likely be correct, but per your statement here, the case would have been excluded? Similarly, if you only included cases where the pathologists agreed, it doesn't make sense to analyze agreement between the 2 clinical pathologists, as they already agree by definition. Often these cavitary infections are polymicrobial. So, I think there is a lot about the materials and methods that needs to be described more clearly/thoroughly.

Reply to R1.11: We appreciate your attention to detail and your constructive suggestions for improvement. We want to clarify that cases were excluded from the analysis only when there was disagreement between the two pathologists regarding the cytological interpretation. This decision was made to assess the concordance between cytology and microbiology findings. However, we included all cases to evaluate the agreement between the two pathologists.

We understand your concern about the potential exclusion of cases where both pathologists may have provided valid interpretations, particularly in polymicrobial infections. We will ensure that our "case definition" of septic versus non-septic effusion is clearly described in the manuscript, including the criteria used to determine agreement between pathologists and the rationale behind the exclusion of cases.

Your feedback underscores the importance of providing thorough and transparent descriptions of our methodology to ensure clarity and reproducibility. We will revise the manuscript accordingly to address your concerns. If you have any further questions or suggestions, please feel free to share them with us. Your input is invaluable to us as we strive to improve the quality and accuracy of our research.

R.1.12: - Line 144: I would eliminate "Antimicrobial Susceptibility" from the title, as you did not report AST in your results, just speciation/culture results (this is perfectly fine, but the focus of your study is on the culture/bacterial isolation, not the AST). 

Reply to R1.12: We agree with the comments of reviewer and have modified the revised version of the manuscript.

R.1.13: - Line 147: is this the methodology just for anaerobic cultures specifically? I would make that clear - something like "The methodology used for anaerobic cultures was..." and in the following paragraph, similarly, introduce that that methodology was for aerobic culture.

Reply to R1.13: We agree with the comments of reviewer and have modified the revised version of the manuscript to avoid misunderstanding.

R.1.14: - Lines 177-182: I would argue this section belongs in section 2.3, not the stats. However, the description in these lines is unclear how you defined certain findings. I recommend clearly stating what test/result you were using as your "gold standard" to compare to (knowing that it is unlikely a true gold standard) and then specific definitions of "positive cytology test" and "true positive cytology test." It's unclear as written right now what the difference is or why it matters.

Reply to R1.14: According to the suggestions and the comments of the reviewer we rephrased the paragraph.  

R.1.15: - Line 196: it's most acceptable to report descriptive statistics such as mean vs. median years accompanied by standard deviation or range respectively. You also never mentioned in your stats section how you were determining if the data for each variable such as these were normally distributed (Shaprio Wilk test? Something similar?). You should also include a statement in your stats section along the lines of "continuous patient data are reported as mean +/- standard deviation when normally distributed and as median (range) when non-normally distributed" or something to that effect. But reporting both the mean and median is not customary.

Reply to R.1.15: To avoid misunderstanding, following the reviewer’s recommendation we used Kolmogorov-Smirnov test and Shapiro Wilk test to verify the normal distribution of the continuous data and continuous variables were summarized as mean and standard deviation when normally distributed and as median and range when non-normally distributed, while categorical data were summarized as count and percentage. We modified the revised manuscript, removing the paragraph indicated by the reviewer, and the last continuous variable is the age of the animals included in the study.

R.1.16: - Table 1: again, a test of normality needs to be performed and reported for these continuous data, and the appropriate descriptive statistic reported. If that was performed and all were normally distributed, report the mean and SD. If non-normally distributed, report the median and range. Another question I have is whether you hypothesize that something about these variables (SG, TNCC, RBC, Ht, TP, LDH) affected the interpretation of cytology as septic vs. non-septic. Otherwise, why report them at all? It would make more sense to me to have certain cutoffs for the fluid analysis as deeming it non-septic (very low TNCC, low LDH, etc.). But as of now, it's unclear why you're reporting the results of the fluid analysis itself when what you aim to compare is the pathologist's cytology results vs. the culture result.

Reply to R1.16: This part was removed.

R.1.17: - Lines 208-213: Why are the results of the gram stain not incorporated into this? Specifically if rods were most common, what is the breakdown of gram positive vs. gram negative rods - those have very different implications when you're comparing to the results of cultures (as an example, if the pathologist read out of cytology says "gram positive rods" and the culture result is E. coli, the pathologist was wrong and those results are discordant, and you should comment on why that might be in the discussion).

Reply to R1.17: Thank you for your insightful comment. Our primary objective in this study was not to compare the results of the Gram stain directly, but rather to enhance the accuracy of diagnosing septic versus non-septic conditions. While the differentiation between gram-positive and gram-negative rods indeed carries significant implications for culture results, it was beyond the scope of our current study. We aimed to compare the septic/non-septic categorization between cytology and microbiology. However, we acknowledge the importance of incorporating Gram stain results and will consider this for a subsequent study. This could provide a more comprehensive understanding and further validate our findings. Thank you for highlighting this point.

R.1.18: - Table 2: it's unclear to me what "Observer 2" indicates when the title of the table implies that these are results from both observers? Are these just the results they agreed upon?

Reply to R1.18: We agree with the reviewer’s observation: we were maybe unclear in reporting what table describes. These are not the results on which the two observers agreed upon. These are all the results recorded by the two observers on the whole caseload, as also specified in the text following figure 1. We have now modified the figure legend to hopefully increase the comprehensibility of the Table (now renumbered as table 1, after removal of the former table 1).

R.1.19: - Figure 1: as you alluded to in your intro, one cannot claim that figure 1 panel A is indeed streptococci, only that it was "gram positive cocci arranged in chains." If the culture result from that same fluid cytology agrees that it is Streptococcus, that's great, but you cannot claim that speciation from the picture alone. 

Reply to R1.19: We agree with the comments of reviewer and to avoid misunderstanding we have modified the figure legend in the revised version of the manuscript.

R.1.20: - Also regarding figure 1/your methods: if bacteria were noted specifically only intracellularly, how did you interpret the cytologic conclusion of a Gram stain? in theory, a Gram stain can only help delineate between gram positive vs. gram negative bacteria when they are extracellular. So for figure 1 panel B for example, how can you conclude that that was gram negative rods, when it should not pick up the Gram stain if it's intracellular bacteria regardless of whether it's truly a gram positive or gram negative rod bacterial species?

Reply to R.1.20: Thank you for your insightful question regarding Figure 1 and our methodology. We appreciate your attention to detail and your interest in understanding our procedures.

Regarding your concern about the interpretation of the Gram stain in the presence of intracellular bacteria, we want to clarify that the Gram stain can indeed detect both intracellular and extracellular bacteria. It effectively stains bacterial cell walls, allowing for the differentiation between Gram-positive and Gram-negative bacteria. Here we report samples of both Gram-positive and Gram-negative bacteria correctly stained inside inflammatory cells:

In our study, when we noted the presence of bacteria specifically intracellularly, we still interpreted the Gram stain results to determine whether the bacteria appeared Gram-positive or Gram-negative. This interpretation is based on the staining characteristics of the bacterial cell wall, regardless of whether the bacteria are intracellular or extracellular.

We understand that certain obligate intracellular bacteria, such as Ehrlichia and Bartonella, may not be visible with the Gram stain due to their unique properties. However, in our study, we focused on bacteria that were phagocytized and visible within the cell, rather than obligate intracellular bacteria.

We hope this explanation clarifies any confusion, and we appreciate your attention to this aspect of our methodology. We add two mixed cytology in the figure 1 of the revised version of the manuscript according to this comment.

R.1.21: - Line 224/225: This sentence as written does not make sense. What are you trying to relay here?

Reply to R.1.21: We agree with the reviewer's comments and have rephrased the sentence in the revised version of the manuscript.

R.1.22: - Table 3/general comment: while some of the organisms you listed such as E. coli may be categorized as a facultative anaerobe, some of the confusion in your introduction and your categorization here may be because most clinicians would consider an organism like E. coli to be an aerobic organism (yes, facultative anaerobe is a subtype of this, but given that you didn't further categorize organisms into microaerophiles, obligate aerobes, aerotolerant anaerobes etc., I would recommend amending your categorizations to strictly "anaerobes" and "aerobes" to limit this confusion. This may vary by geographic location as well, but where I live and practice, if a trainee claimed that an organism like E. coli was "an anaerobe" when what they meant was "facultative anaerobe", I would pronounce them incorrect.

Reply to R1.22: According to the reviewer's kind suggestions and comments, we have modified the revised manuscript.

R.1.23: - Table 5: in all 3 of these comparisons (MGG, gram, either), there are relatively high false negative rates (20.5%, 24.6%, 33.6%) - were these mostly in anaerobic infections? We talk about how anaerobic cultures in theory have higher false negative rates, but in this table, the message I am getting is that there were a good number of samples which were negative for bacteria on cytology but had positive growth - I would recommend commenting on that subpopulation of false negative cytologies and the types of organisms they grew out on culture testing to look for any patterns or commonalities. Conversely, were there cases where both pathologists agreed that bacteria were present in the sample, but the culture result was negative?

Reply to R1.23: The false negative rates reported in table 5 refer to samples that were negative in cytology but had positive bacterial growth (see also the definition of false negative/false positive results that has been added to the material and method section). The source of this relatively high rate of false negative results has been described here in the results section by adding new information on the sentence about the high prevalence of G- bacteria that was already present in the former version of the manuscript. We agree with the comments of reviewer and we have modified the revised version of the manuscript.

R.1.24: - Table 6: given the very few pericardial infections, it should be noted in the discussion that no meaningful conclusions can be made due to too few isolates.

Reply to R1.24: As the reviewer suggested, we included a comment on the pericardial infection results in the discussion. 

R.1.25: - General comment: in addition, since it seems that the majority of your study population was feline pyothorax, that should be discussed as a limitation in your discussion.

Reply to R1.25: As the reviewer suggested, we included a comment on the limitation of our study in the discussion.

R.1.26: - Line 301-302: Looking at the parameters reported in Table 9, I do not think this statement is correct (the 2 observers seem extremely close in all parameters when you look at gram vs. MGG stains separately - for example, Sn of gram stains was 64.2% observer 1 and 65.7% observer 2; PPV for MGG was 96% for observer 1, 95.9% for observer 2, etc.). Again, in your discussion on Lines 326-332 you claim this, which I believe is not correct.

Reply to R1.26: We agree that the differences in the diagnostic performances between observer and stains were minimal. We have reworded the sentence accordingly.

R.1.27: - Line 316: culture is also used to confirm or refute the presence of infection, I would argue, not just to guide antimicrobial therapy.

Reply to R1.27: We agree with the comments of reviewer and to avoid misunderstanding we have modified the revised version of the manuscript.

R.1.28: - I had the impression from your methods description that you were going to assess whether the shape/gram stain criteria on cytology matched the organism identified on culture (e.g. were their gram-positive cocci on cytology which was confirmed as Staphylococcus on culture [concordant result] vs. confirmed as E. coli on culture [discordant result]). I don't see this type of analysis described in the results - if that was not the intention, please clarify the methodology as mentioned in comments above. 

Reply to R1.28: We agree with the comments of reviewer. In the discussion, we should have better clarified this point: the aim of our study was not to evaluate concordance with Gram, but only to see the cytology-microbiology concordance as a first step. A future step could be to evaluate Gram separately and we would like to thank for this precious feedback that opens future prospectives.

R.1.29: - Lines 367-371: Don't you think that the most common isolates for cats were due to the large proportion of pyothorax cases? We do expect typically a different distribution of the frequencies of certain bacterial species depending on the location and thus the source of infection. Many septic effusion studies in dogs have focused on abdominal sepsis, as those cases often are associated with a worse prognosis and often more severe disease compared with canine pyothorax. I suggest commenting on this discrepancy in this area of the discussion.

Reply to R1.29: Thank you for your feedback. We have revised the manuscript in response to your suggestion. Specifically, in lines 367-371, we have addressed the potential influence of differing infection sources on the distribution of bacterial species, particularly in cases of pyothorax in cats. Additionally, we have commented on the discrepancy in the distribution of bacterial species between septic effusion studies in dogs with abdominal sepsis and those with canine pyothorax. We appreciate your valuable input and hope that these modifications align with your expectations.

R.1.30: - Line 394: you focused on culture being the reference diagnostic and comparing cytology results to it, but did not comment much on the limitations of either test alone to substantiate this comment: "...in light of the results obtained it was found that both bacteriology and cytology have limitations especially when taken individually." etc.

Reply to R1.30: Thank you for your thoughtful comment. We appreciate your attention to detail. The points you raised regarding the limitations of culture and cytology as individual diagnostic tests have indeed been addressed in the preceding paragraph (line 410-429), where we discussed the challenges associated with each method separately. We have taken your feedback into consideration and ensured that the limitations of both culture and cytology are adequately acknowledged in our manuscript. As we have already cited these points earlier in the manuscript, we do not plan to make further modifications at this time.

R.1.31: - You have not compared your study design and findings to that of Allen et al. 2022, which I think is warranted.

Reply to R1.31: Thank you for bringing up the comparison with Allen et al. 2022. We acknowledge the importance of situating our study within the context of existing literature, including relevant prior research such as Allen et al. 2022.

We will carefully review their study design, findings, and conclusions to identify similarities and differences with our own research. Incorporating this comparison into our manuscript will provide readers with a more comprehensive understanding of how our study contributes to the current body of knowledge on the topic.

Your suggestion highlights the need to ensure that our study is appropriately contextualized within the broader scientific landscape, and we appreciate your guidance in this regard.

In the revised manuscript, we included a paragraph comparing our study results to those of Allen et al. 2022.

R.1.32: While the text is relatively easy to follow, I do recommend a native English speaker review the manuscript after adjustments to the above comments/concerns are made to increase the clarity of the content.

Reply to R1.32: The manuscript has been extensively revised for the use of the English language.

Reviewer 2 Report

Comments and Suggestions for Authors

The study is fascinating, especially for its practical and clinical aspects. The importance of proper technique for anaerobic culture is crucial and the experience of the operator and the short times between sample collection and arrival at the laboratory should be emphasized.

Author Response

Dear Editor,

thank you for giving us the opportunity to submit a revised draft of our manuscript titled: “Diagnosis of Septic Body Cavity Effusion in Dogs and cats: Cytology Vs Bacterial Culture”, Manuscript ID: animals-2982837, submitted to Animals. We appreciate the time and effort that you and the reviewers have dedicated to providing your valuable feedback on our manuscript. We are grateful to the reviewers for their insightful comments on our paper. We have been able to incorporate changes to reflect most of the suggestions provided by the reviewers. We have highlighted the changes within the manuscript.

Here is a point-by-point response to the reviewers’ comments and concerns.

Comments from Reviewer 2 (R.2)

The study is fascinating, especially for its practical and clinical aspects. The importance of proper technique for anaerobic culture is crucial and the experience of the operator and the short times between sample collection and arrival at the laboratory should be emphasized.

 Reply to R2: We are extremely grateful to the referee for the positive feedback.
